# Feasibility of establishing an infant hearing screening program and measuring hearing loss among infants at a regional referral hospital in south western Uganda

Amina Seguya[1]*, Francis Bajunirwe[2], Elijah Kakande[3], Doreen Nakku[4]

1 Department of Ear, Nose and Throat Surgery, Mulago National Referral Hospital, Kampala, Uganda,
2 Department of Community Health, Mbarara University of Science and Technology, Mbarara, Uganda,
3 Infectious Diseases Research Collaboration, Kampala, Uganda, 4 Department of Ear, Nose and Throat Surgery, Mbarara University of Science and Technology, Mbarara, Uganda

* seguya.amina@gmail.com

**Data Availability Statement:** All relevant data are within the manuscript and its Supporting information files.

## Abstract

### Introduction

Despite the high burden of hearing loss (HL) globaly, most countries in resource limited settings lack infant hearing screening programs(IHS) for early HL detection. We examined the feasibility of establishing an IHS program in this setting, and in this pilot program measured the prevalence of infant hearing loss (IHL) and described the characteristics of the infants with HL.

### Methods

We assessed feasibility of establishing an IHS program at a regional referral hospital in south-western Uganda. We recruited infants aged 1 day to 3 months and performed a three-staged screening. At stage 1, we used Transient Evoked Oto-acoustic Emissions (TEOAEs), at stage 2 we repeated TEOAEs for infants who failed TEOAEs at stage 1 and at stage 3, we conducted Automated brainstem responses(ABRs) for those who failed stage 2. IHL was present if they failed an ABR at 35dBHL.

### Results

We screened 401 infants, mean age was 7.2 days (SD = 7.1). 74.6% (299 of 401) passed stage 1, the rest (25.4% or 102 of 401) were referred for stage 2. Of those referred (n = 102), only 34.3% (35 of 102) returned for stage 2 screening. About 14.3% (5/35) failed the repeat TEOAEs in at least one ear. At stage 3, 80% (4 of 5) failed the ABR screening in at least one ear, while 25% (n = 1) failed the test bilaterally. Among the 334 infants that completed the staged screening, the prevalence of IHL was 4/334 or 12 per 1000. Risk factors to IHL were Newborn Special Care Unit (NSCU) admission, gentamycin or oxygen therapy and prematurity.

**Funding:** This research was funded by a small grant from the Brigham and Women's Hospital, Department of Otolaryngology, Harvard medical School to the Department of ENT surgery at Mbarara University of Science and Technology to support postgraduate research for AS. The funders had no role in study design, data collection and analysis, decision to publish, or preparation of the manuscript.

**Competing interests:** The authors have declared no competing interests.

## Conclusions

IHS program establishment in a resource limited setting is feasible. Preliminary data indicate a high prevalence of IHL. Targeted screening of infants at high risk may be a more realistic and sustainable initial step towards establishing IHS program s in a developing country like Uganda.

## Introduction

Hearing loss (HL) is the fourth highest ranked cause of disability globally [1] and the World Health Organisation (WHO) estimates that up to 2.5 billion people globally will have some degree of hearing loss by the year 2050 [2]. Early identification and timely intervention have been promoted by the WHO [3] and The Joint Committee on Infant Hearing [4] as important action points for slowing down the fast rising numbers of disabling Infant Hearing Loss (IHL). However, despite the higher prevalence of IHL reported in Low and Middle Income Countries (LMICs) [5] as compared to high income countries (HIC) [1,6], progress towards achieving early identification of IHL in the LMICs has been slow [7,8].

Infant hearing screening (IHS) is a globally accepted and promoted method for early detection of infants with HL [1,4] in order to facilitate application of timely interventions [9]. Early interventions for HL enable proper cognitive, psychological, speech and language development among the affected infants [4,10]. IHS is performed using either Transient Evoked Oto-acoustic Emissions (OAEs), and or Automated Auditory Brainstem responses (ABR) to record a physiologic response from the ear [4,11,12]. The advantage with these tests is that they are quick, objective and easy to administer yet effective in identifying infants with HL [13].

Although many high income countries have well developed and successful IHS programs targeting all infants [2,14] such programs are lacking in majority of the LMICs [15,16]. The barriers to implementation of these programs in LMICs include non-prioritization of the services, limited resources to purchase the expensive equipment required and lack of expertise to perform and interpret the tests among others [16–19]. In Uganda, establishment of infant hearing screening and rehabilitation programs has been attempted on a small scale in a few centers but has not taken root with reports showing that less than 1% of all newborns in Uganda are screened for hearing loss [14].

The majority of countries that lack IHS programs are in resource limited settings [14,16] and need motivation to establish the programs and initiate services for screening. Therefore, there is a need for data and shared experiences on the feasibility of establishing infant hearing screening programs from resource limited settings. These data will provide lessons for health workers and policy makers to replicate. The programs will also provide data on the prevalence of infant hearing loss and inform strategies for scaling these screening programs to ensure they reach those who need them the most. Therefore, the purpose of this study was to determine the feasibility of establishing an infant hearing screening program in a resource limited setting, measure hearing loss prevalence and describe the characteristics of the infants with hearing loss.

## Materials and methods

### Study design and setting

We conducted the feasibility study at Mbarara Regional Referral Hospital (MRRH), a large tertiary referral hospital that doubles as a regional referral hospital and a university teaching

hospital for Mbarara University of Science and Technology's Faculty of Medicine, in south-western Uganda. The hospital serves as a catchment of approximately 4 million, which is pre-dominantly from rural populations. The facility is the only center that offers specialized care for Ear Nose and Throat (ENT) and Audiology services in southwestern Uganda. The hospital lacks a formal neonatal Intensive care unit (NICU), but all sick newborns receive care in a newborn special care unit (NSCU) housed on the pediatric ward.

### Establishment of screening program: Background and stakeholder engagement

We identified and engaged the key stakeholders in the hospital including hospital administration, the clinical staff of the Department of Pediatrics and Child Health and the Department of ENT which houses the Audiology division. We held several discussions about infant hearing screening with the hospital administration to sensitize them and requested for space and the basic requirements needed to establish the infant hearing screening services at the hospital.

### Equipment acquisition and IHS team development

Since the hospital did not have the necessary equipment required to perform the hearing screening tests, we applied for a donation of the test equipment from a private audiology center to use their PATH Sentiero Medical diagnostics desktop devices; one for TEOAEs testing and another for ABRs testing. We also requested for necessary items including disposable skin electrodes, antiseptic for cleaning the equipment and the infant ear tips for various ear canal sizes.

We established an IHS team comprised of a research assistant, a pediatric nurse, an ENT resident and an ENT surgeon with additional audiology training. The team received training in Infant hearing loss and infant hearing screening using TEOAs and ABRs testing with the donated equipment by the ENT surgeon over two weeks prior to study commencement. The screening tests for this pilot study were performed by the Principal Investigator (AS) under supervision of the ENT surgeon (DN).

### Hearing screening procedure

Hearing screening was performed using TEOAEs and ABRs as recommended by the joint committee on infant hearing [4] following a multi-staged screening format which has the advantage of reducing the referral rate for diagnostic audiometric testing when compared to single stage screening [20]. This multiple staged screening comprised of three stages; stage 1 which used TEOAEs, stage 2 involved repeat TEOAEs for infants that failed stage 1 and at stage 3, we performed ABR for infants that failed stage 2.

At stage 1, appropriately sized ear tips were inserted into the infant's ear, click stimulus introduced in the test ear and TEOAEs recorded with 3 possible results; 'pass' (equivocal to normal hearing) 'fail' (equivocal to likely hearing loss) or 'inconclusive' (if test result was unclear or test was not completed after 2 attempts). Infants with a fail or inconclusive result at stage 1 in at least 1 ear were referred to stage 2 for repeat TEOAE screening for the procedure to be done after two weeks or at their next outpatient clinic appointment, while those that passed the stage 1 test in both ears were discharged from the study with a written record of their test result.

At stage 2, both ears were re-screened with TEOAEs, and if a 'pass' result for both ears (equivocal to normal hearing) was attained, the infant was discharged from the study with a written record of their result. However, a second time fail or inconclusive result in any ear was followed by stage 3 screening with an air conduction ABR test at 35dB HL performed during the same visit. Adjustable headsets placed on the ears and electrodes placed on the forehead

and mastoid bones. A sound chirp stimulus at 35dB HL was introduced into the test ear and an ABR wave generated by the test machine. Infants with a normal wave were considered as a 'pass' (equivocal to normal hearing) and were discharged from the study with a written record of their results. Infants with either abnormal or no ABR wave pattern at 35dBHL failed the test and were considered to have hearing loss. All tests were performed in a quiet room by the ward or clinic or in a sound treated audiology room.

## Eligibility criteria

Based on the guidelines provided by the Joint Committee on Infant Hearing (JCIH) to diagnose infant hearing loss by 3 months of age [4], we recruited infants aged 1 to 3 months of age. Infants were broadly grouped into 2 categories namely the healthy and sick infants. The healthy infants were recruited from the immunization and well child clinics and these included symptom free infants that were not receiving treatment for any disease or condition. We recruited the sick infants from among those who were admitted to the newborn special care unit (NSCU) and these infants were enrolled into the pilot program at discharge from the NSCU after completing in-patient treatment for their various admission diagnoses. The mothers provided written informed consent to participate in the study before enrolment of the children. We excluded infants whose biological mothers were absent and those with craniofacial anomalies that made it difficult to adequately fit the test equipment.

## Sampling, sample size calculation and data collection

Using the Kish Leslie formula for descriptive analysis [21], we calculated a minimum sample size of 384 infants as the minimum number of infants required to assess prevalence of IHL

In a stratified approach, we separately enrolled the unwell and healthy infants with the aim to enroll an equal number of infants from each category. We enrolled the unwell children at their discharge from the NSCU. We made the assumption that there is no particular pattern in the order of discharge. We applied a systematic sampling approach to select the eligible infants until the required sample size was achieved. We recruited the healthy infants from the immunization and well child clinics and enrolled every third infant that visited the clinic. In the event that a mother declined to provide consent, the next eligible infant was considered.

We interviewed mothers using a semi-structured tool to collect data regarding socio-demographic, medical and perinatal details. Data on presence of risk factors to infant hearing loss and the hearing screening results were recorded for each ear in the data collection tool. We collected data between July 2018 to March 2019.

## Data analysis

Data were analyzed using STATA version 14.0 as the statistical data analysis software. Descriptive statistics were used to describe the socio demographic characteristics of the mothers and infants and the risk factors for HL among the infants and these were presented as frequencies. The primary outcome from this pilot study was infant hearing loss. We defined presence of infant hearing loss if the infant failed the TEOAEs screening twice and failed an ABR at 35dB HL in at least on ear. The prevalence of IHL was calculated as the number of enrolled infants who completed staged screening and failed TEOAEs and ABR testing in at least one ear over the total number of infants that completed staged screening. We excluded were unable to complete all the stages of screening as required.

We assessed the feasibility of establishing an infant hearing screening program as based on key stake holder buy in and the success to: acquiring the required space for performing the hearing screening tests, acquiring and training the required human and technical support

from the various key stakeholders in the hospital, acquiring the necessary auditory equipment required for screening of the infants and the acceptance of mothers to provide consent for their infants to have hearing screening performed.

### Ethical considerations

Ethical clearance was received from the Mbarara University of Science and Technology Research Ethics Committee (IRB no.05/04-18) before commencement of the study. The risks and benefits of the study were explained to the mothers and only those mothers who provided written informed consent to have their children to participate in the study were enrolled. For the mothers whose children were identified as having hearing loss, the results were explained to the mothers and the infants were referred to the ENT clinic for further assessment and work up for rehabilitation.

## Results

### Feasibility of establishing an IHS program

We obtained buy in from the hospital administration and acquired the necessary permissions and resources to support initiation of Infant hearing screening at the hospital. The Hospital administration allocated us the required space for conducting the study. We received two rooms, one in the NSCU and another in the pediatric wellness clinic. These rooms were used for conducting interviews with the mothers plus performing the hearing tests. The Hospital also granted us permission to use the audiology room in the ENT department, which did not need any modifications since it is already sound proofed and suitable for audiology tests like ABRs that require a quiet room.

We also received a donation of equipment to perform TEOAEs and ABRs plus the necessary consumables for the duration of the study. This contributed immensely to the success of the study because our tertiary center did not own equipment to perform IHS.

We successfully formed an IHS team and this was comprised of a research assistant, a pediatric nurse from the pediatric ward, an ENT resident and an ENT surgeon with additional audiology training. The team was successfully trained in IHS screening for a duration of two weeks. This team successfully performed the IHS screening for the duration of the study.

Although majority of the mothers had never heard about IHS, mobilizing mothers following a brief explanation about IHS and IHL was smooth. Mothers approached were mostly positive about knowing the status of hearing in their infants. 28 mothers declined consent for their infants to be screened and they cited fear of long term or unseen effects of the IHS equipment on a newborn, having no previous knowledge about IHS and absence of the father of the infants to consent to the test and study participation as reasons for decline to consent for IHS.

### Sociodemographic characteristics of the infants

A total of 401 infants were recruited into this study. We enrolled 203 infants (50.6%) from the NSCU and 198 healthy infants or 49.4% from the pediatric immunisation and wellness clinics or as shown in Table 1. The mean age was 7.2 days (standard deviation = 7.1 days). Majority of the infants in the study were male (56% or 255 of 401), with no significant variation by gender between the infants fron NSCU and the healthy infants.

Although majority of the infants in the study were delivered in a health centre, 19% (76 of 401) were delivered elsewhere, with more infants in the NSCU group delivered away from a health centre (8.4% or 17 of 203) as compared to the healthy infants (2% or 4 of 196).

**Table 1. A table showing the demographic characteristics of the infants (n = 401) enrolled in a hearing screening program at Mbarara Regional referral hospital.**

| Variable | All Infants N = 401 n(%) | Sick babies N = 203 n(%) | Healthy babies N = 198 n(%) | p value |
|---|---|---|---|---|
| **Gender** | | | | |
| Male | 255 (56.1) | 112 (55.2) | 113 (57.1) | 0.684 |
| Female | 176 (43.9) | 91 (44.8) | 85 (42.9) | |
| **Birth weight** | | | | |
| >2.5kg | 300 (74.8) | 121 (59.6) | 170 (90.4) | **<0.001*** |
| 1.5–2.49kg | 83 (20.7) | 64 (31.5) | 19 (9.6) | |
| <1.5kg | 18 (4.5) | 18 (8.9) | 0 (0) | |
| **Family history of hearing loss** | | | | |
| No | 297 (74.1) | 160 (78.8) | 137 (69.2) | **0.002*** |
| Yes | 104 (25.9) | 43 (21.2) | 61 (30.8) | |
| **Delivery in health center** | | | | |
| No | 76 (19.0) | 17 (8.4) | 4 (2.0) | **<0.005*** |
| Yes | 325 (81.0) | 186 (91.6) | 194 (98.0) | |
| **Report of immediate cry at birth** | | | | |
| No | 76 (19.0) | 63 (31.0) | 13 (6.6) | **<0.001*** |
| Yes | 325 (81.0) | 140 (69.0) | 185 (93.4) | |
| **Oxygen therapy** | | | | |
| No | 318 (79.3) | 122 (59.5) | 196 (0) | **<0.001*** |
| 1–5 days | 79 (29.7) | 79 (38.5) | 0 (0) | |
| More than 5 days | 4 (1.0) | 4 (2.0) | 0 (0) | |
| **Gentamycin therapy** | | | | |
| No | 208 (51.9) | 12 (5.9) | 196 (100.0) | **<0.001*** |
| 1–5 days | 165 (41.1) | 165 (80.5) | 0 (0.0) | |
| More than 5 days | 28 (7.0) | 28 (13.6) | 0 (0.0) | |

*p< than 0.05.

For an additional 19% or 76 of 401 infants, no cry was reported at birth. Majority of these infants were in the NSCU category (31% or 63 of 203) as compared to 6.6% or 13 of 196 infants.

None of the healthy infants received oxygen or gentamycin therapy, while among the NSCU infants, 80.5% (165 of 203) and 13.6% (28 of 203) received gentamycin therapy for 1 to 5 days and more than 5 days respectively and 38.5% (79 of 203) and 2% (4 of 203) received oxygen therapy for 1 to 5 days and more than 5 days as shown in Table 1.

Among the babies recruited from the NSCU (n = 205), the commonest admission diagnosis was neonatal sepsis in 25.9% (n = 53) followed by prematurity at 19% (n = 39) and birth asphyxia at 11.2% (n = 23). Other diagnoses at admission included jaundice among 14.6% (n = 30), birth asphyxia among 11.2% (n = 23), admission for surgery among 4.4% (n = 9) and meningitis among 1% (n = 2). Significantly only 1% (4 of 401) infants had received oxygen therapy for more than 5 days as shown in Table 1.

## Hearing screening results

Of the 401 infants that successfully completed stage 1 TEOAEs screening, 299 of 401 (74.6%) passed while at least one quarter (102 of 401 or 25.4%) of the infants had either an inconclusive or fail test result and were referred to stage 2 for a repeat TEOAEs screening as shown in Fig 1. However only 35 of the 102 infants or 34.3% returned and had stage 2 repeat screening

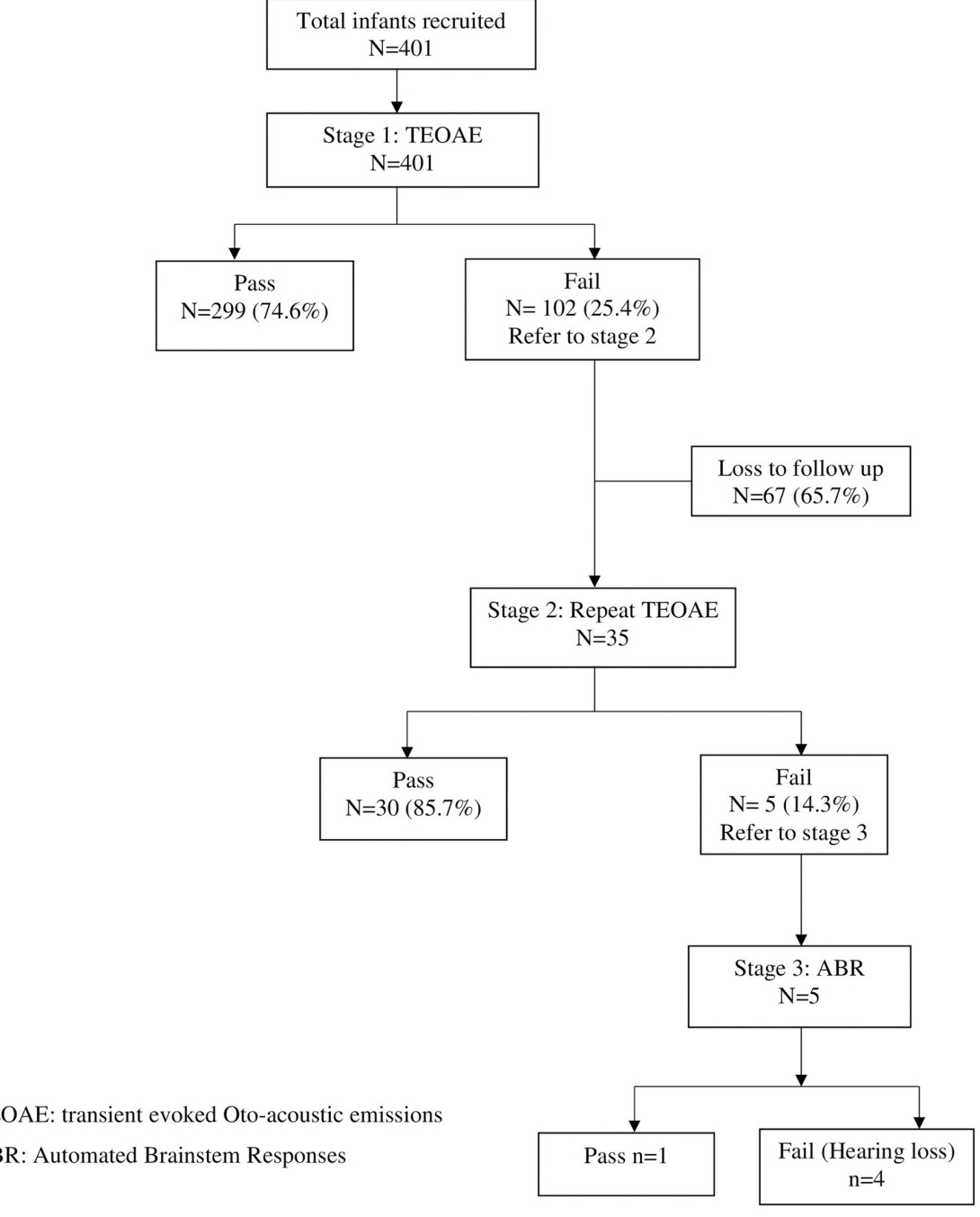

**Fig 1. A summary of results of infant hearing screening using TEOAE and ABR tests.**

and two thirds (67 or 65.7%) failed to return for repeat screening. Of the 35 infants that returned for stage 2 screening, 5 of 35 (14.2%) infants failed the test in at least one ear and proceeded to stage 3 for ABR screening. Of these 5 infants, one subsequently passed the stage 3 ABR, one failed in both ears and 3 failed in at least one ear. A total of 334 infants completed all the stages of screening as was required, yielding a prevalence of hearing loss of 4/334 or 12/1000.

**Table 2. A figure showing distribution of risk factors among the infants with hearing loss.**

| Infants with hearing loss | Risk factors present |
|---|---|
| Infant 1 | NSCU admission,sepsis, jaundice, oxygen therapy, gentamycin therapy, syphillis infection |
| Infant 2 | NSCU admission, prematurity, juaandice, oxygen theray, gentamycin therapy |
| Infant 3 | NSCU admission, prematurity, gentamycin therapy |
| Infant 4 | NSCU admission, osteogenesis imperfecta syndrome, oxygen therapy, gentamycin therapy |

NSCU- Newborn special care Unit.

## Risk factors for HL among infants

We found that all the infants diagnosed with hearing loss (n = 4) presented with at least 4 of the known risk factors for hearing loss while one infant had up to 7 factors present as represented in Table 2. Admission to NSCU was the commonest risk factor and was present in all the infants with HL.

Other risk factors for hearing loss that were present among the infants with HL included prematurity, sepsis, osteogenesis imperfecta, jaundice, oxygen therapy and Gentamycin therapy as shown in Table 2.

## Discussion

Our experience shows that establishing an infant hearing screening program in a resource limited setting such as that at a tertiary hospital in Uganda is feasible. This was the first effort to initiate an IHS program at a referral hospital in the south western region of Uganda and IHS was performed successfully at this tertiary center for 9 months. The success may be attributed to the positive responses and support from the key stakeholders in this tertiary hospital.

We successfully managed to perform infant hearing screening for the duration of the study with positive support from stake holders including the hospital administration, hospital departments and mothers. The hospital administration was very forthcoming in providing the necessary resources where available such as space, human resource and some equipment required for performance of IHS. Mothers were keen to have their babies tested and even after recruitment, several others continued to request for the service. This gave us the impression that this is a much needed service and if rolled out would be embraced and possibly integrated as part of the routine care for all infants with in the hospital.

Although we found it feasible to perform IHS at this tertiary center, the initial challenges we faced included lack of human resource trained in IHS and lack of equipment to perform the IHS at the hospital. We overcame these challenges by creating an IHS team and providing essential IHS training to these team members. Since the hospital did not own IHS testing equipment, we requested and successfully received a donation of IHS equipment from a nearby audiology center. Similar challenges have been faced by other IHS programs in LMICs, with limited human resource both in numbers and training reported in both South Africa [22] and in Pakistan [18] while lack of appropriate audiology equipment was similarly reported in South Africa [17].

The acceptance rate to participate in the screening was high. The few mothers (28 mothers) that declined to participate in this study cited fear of the unfamiliar infant hearing screening technology, having no previous knowledge about infant hearing loss and absence of the decision making parent/guardian as reasons for refusal to consent. Similarly high rates of

acceptance of infant hearing screening have been reported by Olusanya et el in Nigeria [23] and Swanepoel et al in South Africa [24]. Although majority of the mothers were very receptive of the service, the reasons given by those that declined should not be neglected if IHS programs are to be successful. Another commonly encountered challenge is ambient noise which affects the test quality [22] however, we overcame this through use of either a quiet room and audio booth where necessary.

We experienced significant loss to follow up of the infants that were required to return for repeat screening. This was probably the biggest challenge in this pilot program and not surprising that it has been reported as a major barrier to early diagnosis of IHL and impacts success of any IHS program [22,25]. The magnitude of infants lost to follow-up or not re-tested varies with higher numbers reported for LMICs such as ours, India (64.6%) [26], Nigeria (51.6%) [5] and Cote d'Ivoire (82%) [27] while lower figures in high income countries like France (7.7%) [28]. We were not able to reach the mothers lost to follow up to obtain the reasons for not returning. However, among the studies conducted in developing countries, the common reasons cited for loss to follow up include financial limitations, fear of spousal reactions and lack of knowledge on deafness as reported in Cote d'Ivoire [27] while long distances to test sites, death of infant and parental refusal for hearing screening were cited in Albania [29].

While the prevalence of IHL in HICs such as Germany [30] and the USA [31] is low, much higher prevalence has been reported in many LMICs as we found in our pilot program. For instance, studies in Cote d'Ivoire have reported a prevalence of 5.96/1000 [27], Jordan was 13.7/1000 [32], Nigeria was 28/1000 [5] and India (6.67/1000) [33]. The high prevalence of IHL in many developing countries may be due poorer obstetric care and early neonatal care in these countries, factors that increase the risk of IHL. The prevalence in our study may be an underestimate because of loss to follow up of some infants that might have had IHL.

Several risk factors for IHL have been reported in literature [24,34–36] but their prevalence varies from region to region. All the infants with hearing loss in our study had multiple risk factors for IHL with admission to NSCU and ototoxic drug administration as the most common factors. Other factors that have been described in literature that were present among these infants included prematurity [34,37,38], jaundice [33,39], and congenital syndromes [34]. Presence of multiple risk factors in an infant has been reported to increase the risk of disabilities such as hearing loss [34,40]. Although birth asphyxia [33,39,41] and neonatal meningitis [37,42] are commonly reported as risk factors for IHL in other areas, these were not present among any of the infants with hearing loss in our study.

For sustainability of an IHS program, it is important for the hospital to acquire the hearing screening equipment so that the trained staff can continue providing these services to the infants. The tertiary hospital already has an audiology department that has a sound proof audio room which can be used room for performing these IHS tests. The ENT department also has post graduate students, an audiologist and an ENT nurse who have received training in performing Infant hearing screening tests from this study. Nurses in the healthy infant clinics and the NSCU can also be trained on Infant hearing screening, and should be part of the IHS team. Formation and continued training of the IHS team is a key step in ensuring sustainability of the infant hearing screening program once the hospital acquires the necessary equipment for infant hearing screening. Education of mothers that attend antenatal care, delivering at the hospital and those that bring their infants to the infant clinics may be a key step in ensuring success of IHS programs. This will dispel any myths or misconceptions about IHS and may reduce loss to follow up.

Rolling out IHS programs for all infants may not be feasible or sustainable at the start, and instead targeted screening of high risk infants may be a key starting point for success of an IHS

program [16]. This focused approach is workable and more cost effective solution for LMICs in which case hearing screening tests are initially availed to only high risk infants [16,35,43–46], until when resources are available for a fully-fledged IHS program for all infants. Targeted infants include those with risk factors such as NSCU admission, ototoxicity, prematurity, low birth weight, mechanical ventilation, hyperbilirubinemia and severe birth asphyxia among others [34–36,41,47].

Our evaluation of the pilot program on infant hearing screening has important lessons and strengths. It is among the first to describe a comprehensive experience of establishing an infant hearing screening program in a low resource hospital setting and also the first to measure the prevalence of IHL in Uganda. We share our experiences that provide important insights for other clinicians in similar settings that have interest to replicate the same program. However, it is not without limitations. First, a large percentage of infants who failed the initial screening did not return for repeat screening and the large percentage of infants lost to follow up could have potentially affected the prevalence reported. Secondly, we screened a small number of infants and hence the prevalence may not be representative of the much larger infant population in Uganda.

We recommend that larger assessments of IHL be performed in Uganda to determine more accurate values for the prevalence of IHL and that there is need for increased community education to promote awareness on Infant hearing loss, infant hearing screening with a lot of emphasis on the importance of the return for rescreening if it is recommended. We also recommend establishment of hearing screening programs in various regional referral hospitals across the country since our study has shown that this is feasible.

## Conclusions and recommendations

Our experience in Uganda shows that establishment of an infant hearing screening program is feasible in a resource limited setting. The prevalence of IHL in Uganda is high with infants admitted to the NSCU being most affected. Although the pilot program was successful, there is concern about the significant loss to follow up after the first screening test, human resources, sustainability and integration of the services into the routine ENT services. Future programs should consider ensuring there are public health awareness campaigns on hearing tests, establishment of multiple screening sites across the country and interventions to reduce loss to follow-up. Due to limited resources for screening and high risk for IHL among NSCU, we recommend targeted screening of high-risk infants for more efficiency.

## Supporting information

**S1 Data.**
(DTA)

## Acknowledgments

We acknowledge and are grateful for the support from the study participants, members in the Departments of Pediatrics and Child Health, Obstetrics and Gynecology and ENT as well as all the staff of Mbarara Regional Referral Hospital. We appreciate our partner, Atlas Audio and Medical Limited for generously loaning us the audiology equipment and providing technical support for the duration of the study.

## Author Contributions

**Conceptualization:** Amina Seguya, Doreen Nakku.

**Data curation:** Amina Seguya, Elijah Kakande.

**Formal analysis:** Elijah Kakande.

**Funding acquisition:** Doreen Nakku.

**Investigation:** Amina Seguya, Doreen Nakku.

**Methodology:** Amina Seguya, Francis Bajunirwe, Elijah Kakande, Doreen Nakku.

**Project administration:** Amina Seguya, Doreen Nakku.

**Resources:** Amina Seguya.

**Software:** Elijah Kakande.

**Supervision:** Francis Bajunirwe, Doreen Nakku.

**Validation:** Elijah Kakande, Doreen Nakku.

**Visualization:** Amina Seguya.

**Writing – original draft:** Amina Seguya.

**Writing – review & editing:** Amina Seguya, Francis Bajunirwe, Elijah Kakande, Doreen Nakku.

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
