## [Decision Letter · Decision Letter 0]

3 Jun 2021

Feasibility of establishing an infant hearing screening program and measuring hearing loss among infants at a regional referral hospital in south western Uganda

PONE-D-21-12598

Dear Dr. Seguya,

We’re pleased to inform you that your manuscript has been judged scientifically suitable for publication and will be formally accepted for publication once it meets all outstanding technical requirements.

Kind regards,

Ricardo Q. Gurgel, PhD

Academic Editor

PLOS ONE

Reviewers' comments:

Reviewer's Responses to Questions

**Comments to the Author**

1. Is the manuscript technically sound, and do the data support the conclusions?

Reviewer #1: Yes

Reviewer #2: Yes

2. Has the statistical analysis been performed appropriately and rigorously? 

Reviewer #1: Yes

Reviewer #2: Yes

3. Have the authors made all data underlying the findings in their manuscript fully available?

Reviewer #1: Yes

Reviewer #2: Yes

4. Is the manuscript presented in an intelligible fashion and written in standard English?

Reviewer #1: Yes

Reviewer #2: Yes

5. Review Comments to the Author

Reviewer #1: As a pilot program, the methodology of the present study allows conclusions around a single health unit, in this case, in a tertiary referral hospital, when in a ideal screening situation would be to do this service in primary care.

I highlight the motivation and altruism of the professionals to proceed and make feasible the screening taking a positive impact for the lives of these children being important tools to others major studies.

During the methodology I had difficulty understanding the steps and premises chosen for the sample size calculation. In addition, as well declared by the authors, we note that the loss of follow-up of the patients between the first and the second stage was high (65.7%) which weakens an external validity of the primary outcome. Considering the loss of follow-up was not possible to reach the "n" predicted at the beginning of the research (384 infants).

However, considering the objective of evaluating the initial feasibility of screening, we should not consider this bias as disruptive in the process of crediting this article. I see with good eyes the fulfillment of the goal of research and that others should bring new solutions, such as the reduction of the loss of follow-up and its transition from a tertiary hospital to a primary care service.

Note:

The 4th paragraph of the Discussion cites that few mothers (less than 1%) declined to partipate in this study but in the Results (4th paragraph) mentioned that they were 28, considering the total of the infants evaluated and the refusals, this would be 6.5%.

Reviewer #2: You've done a good job, despite the quite small sample, following the correct and recent hearing screening guidelines. The references are recent, the statistical analysis is good and you have used a fluent English.

6. PLOS authors have the option to publish the peer review history of their article (what does this mean?). If published, this will include your full peer review and any attached files.

Reviewer #1: No

Reviewer #2: **Yes: **Elisabetta Genovese

---

## [Editor Report · Acceptance letter]

9 Jun 2021

PONE-D-21-12598 

Feasibility of establishing an infant hearing screening program and measuring hearing loss among infants at a regional referral hospital in south western Uganda 

Dear Dr. Seguya:

I'm pleased to inform you that your manuscript has been deemed suitable for publication in PLOS ONE. Congratulations! Your manuscript is now with our production department. 

Kind regards, 

on behalf of

Professor Ricardo Q. Gurgel 

Academic Editor

PLOS ONE